# Bioaccumulation of Potentially Toxic Elements in *Tilia tomentosa* Moench Trees from Urban Parks and Potential Health Risks from Using Leaves and Flowers for Medicinal Purposes

Miroslava Mitrović *[ID], Olga Kostić [ID], Zorana Miletić [ID], Milica Marković [ID], Natalija Radulović [ID], Dimitrije Sekulić [ID], Snežana Jarić [ID] and Pavle Pavlović [ID]

Department of Ecology, Institute for Biological Research 'Siniša Stanković'—National Institute of the Republic of Serbia, University of Belgrade, Bulevar despota Stefana 142, 11060 Belgrade, Serbia; olgak@ibiss.bg.ac.rs (O.K.); zorana.mataruga@ibiss.bg.ac.rs (Z.M.); milica.markovic@ibiss.bg.ac.rs (M.M.); natalija.radulovic@ibiss.bg.ac.rs (N.R.); dimitrije.sekulic@ibiss.bg.ac.rs (D.S.); nena2000@ibiss.bg.ac.rs (S.J.); ppavle@ibiss.bg.ac.rs (P.P.)
* Correspondence: mmit@ibiss.bg.ac.rs; Tel.: +381-11-2078362

**Abstract:** Potentially toxic element (PTE) contamination in medicinal plants, particularly those growing in urban environments, can cause human health issues. Therefore, this study evaluated trace element accumulation and translocation patterns (As, Cr, Cu, Ni, Pb, Sr, and Zn) in the aboveground tissue of common *Tilia tomentosa* Moench, often used as a medicinal plant, sampled in Belgrade's urban parks (Zemunski Park, Park Blok 63, and Park Topčider). Our results indicated that this species exhibits the ability to accumulate and translocate PTEs, particularly Cu, in its aboveground parts. It was found that the levels of Cu and Sr in flowers were within the toxic range for plants, indicating a potential risk in using *T. tomentosa* flowers from Park Topčider for medicinal purposes. The maximum Estimated Daily Intake of Ni from the consumption of leaves and flowers of plants growing in two parks (Zemunski Park and Park Topčider) exceeded the corresponding Provisional Tolerable Daily Intake. Additionally, the Carcinogenic Risk calculated for Cr present in flowers was above the USEPA limit ($3.021 \times 10^{-3}$), indicating possible adverse effects on human health and a carcinogenic risk from ingesting tea prepared from *T. tomentosa* flowers from Park Topčider. Our research underlines how crucial it is to cautiously use medicinal tree species growing in urban parks in residential areas.

**Keywords:** *T. tomentosa*; PTE accumulation; bioconcentration; translocation; medicinal use; health risk




## 1. Introduction

Increasing urbanisation has led to pollution becoming one of the most significant issues in cities, with significant environmental degradation occurring as the population rises. Pollutant emissions in urban areas have a major detrimental effect on human health [1–3]. In particular, air pollutants have become a significant global threat, and currently, almost half of the world's population is exposed to poor air quality [4]. In addition, recent data show that only about 10% of the world's population lives in areas where safe air quality standards set by the World Health Organization are met [5,6]. The contamination of soil also has a negative impact on many of the United Nations' Sustainable Development Goals. These include good health and wellbeing, sustainable ecosystems and cities, and climate change regulation. When occurring in cities, this contamination is mostly linked to vehicle emissions, industry, and poor waste management [7–9]. These issues affect not only individuals, but also medical systems and ecosystem health and disrupts economies, both in developing and in developed countries.

Urban parks are important elements of the urban infrastructure as they improve the environmental quality by providing various ecosystem services, such as the regulation of air, water, and soil pollution and of climate and the provision of habitat quality and of goods and services [10–12]. However, many urban green spaces are designed primarily

for aesthetic appeal, with limited consideration for the broader ecosystem services they can provide [10]. While urbanisation clearly has a negative impact on health, doubts remain over whether green spaces such as parks and playgrounds actually do provide the health benefits that have been reported [13]. There is general agreement that urban trees play a central role in improving urban habitat quality by facilitating the deposition of air pollutants and particles, as they provide large surface areas as well as the uptake of potentially toxic elements (PTEs) from the soil [3,12,14–19].

Trees of the genus Tilia, commonly known as linden trees, are frequently planted in urban green areas and are one of the most popular tree species in urban parks across Europe [20–25] and in Serbia [26–29]. They are cultivated as individual or group trees, as natural avenue trees, or as topiaries in urban parks and along roadsides.

Linden trees are valuable as biomonitors because they accumulate significant amounts of chemical elements, including potentially toxic elements (PTEs), reflecting the extent of pollution in urban ecosystems [26,27,29,30]. In addition, linden is used for the phytoremediation of sites that have been impacted by anthropological activities [31,32]. Therefore, the ability of its roots, leaves, and flowers to accumulate PTEs from either soil or air should be analysed, as its leaves and flowers are used both as raw material and as components in various medicinal remedies including powders, infusions, and extracts [33]. Infusions of Tilia are widely used in traditional medicine in Europe and Latin America for the treatment of gastrointestinal and respiratory ailments, as well as for their calming properties [34–37]. In Serbia, the linden flower is used to treat catarrh and for its diuretic, expectorant, and antispasmodic properties, while the leaf is used as a diaphoretic [38]. Linden is certified as a food product and medicine for the treatment of cold symptoms such as sore throats, coughs, and fevers [39]. It is widely consumed as a food supplement across the world, including in Serbia, especially in the form of linden blossom tea. Due to the scarcity of natural deciduous forests with linden trees, Serbian communities have historically harvested linden leaves and flowers in suburbs and urban parks during the spring and early summer months. Therefore, *T. tomentosa* Moench, was selected for this study because it is a common urban tree species that is used for street planting, urban parks, and green areas surrounding industrial centres due to its high resistance to drought, dust, and industrial pollution [26–28,40]. In this regard, the objectives of this research were to (1) assess the content of potentially toxic elements (PTEs) in the soil in three urban parks in Belgrade, Serbia, (2) assess the ability of *Tilia tomentosa* Moench trees to accumulate and translocate PTEs by measuring PTE concentrations in roots, leaves, and flowers, (3) assess the potential health risks of consuming leaves and flowers based on calculations of the Estimated Daily Intake (EDI), Daily Intake of Metals (DIM), Total Hazard Quotient (THQ), and Carcinogenic Risk (CR), (4) raise residents' awareness that potentially high concentrations of certain PTEs in *T. tomentosa* leaves and flowers may make the continued use and consumption of these products dangerous for health. It is expected that the results of this work will be useful for determining the allocation patterns of PTEs in urban trees as well as for protecting the health of local communities.

## 2. Materials and Methods

### 2.1. Study Areas and Sampling Procedure

For this study, field sampling was conducted in July 2021 at three sampling sites located within residential areas of the city of Belgrade (Serbia): Zemunski Park (ZP) (44°50′23.4″ N, 20°24′35.6″ E), Park Blok 63 (PB63) (44°48′21.6″ N, 20°22′53.6″ E), and Park Topčider (PT) (44°46′55.7″ N, 20°26′21.4″ E) (Figure 1). Park Topčider is one of the oldest parks in Belgrade and was built at the same time as that of a residential complex. It extends over an area of approximately 13 ha and is covered with deciduous and evergreen trees. In 2015, Belgrade City Council passed a decree declaring Park Topčider a Nature Park. Zemunski Park is located in the central area of the municipality of Zemun, covering an area of 7.72 ha, and is bordered by numerous roads. It is also covered with deciduous and evergreen trees. Park Blok 63 is a park that extends over 2.3 ha and is within a large

residential block in the municipality of Novi Beograd. The sampling was conducted in the summer, during *T. tomentosa* flowering season. According to data from the Republic Hydrometeorological Service of Serbia, the mean annual temperature was 13.8 °C, the annual sum of precipitation was 795.3 mm, and thee mean annual air humidity was 67%, while the climatic conditions during the sampling period were as follows: average temperature of 26.6 °C, average precipitation of 63.1 mm, and average air humidity of 57% (https://www.hidmet.gov.rs/data/klimatologija/latin/2021.pdf (accessed on 20 October 2023)). [41].

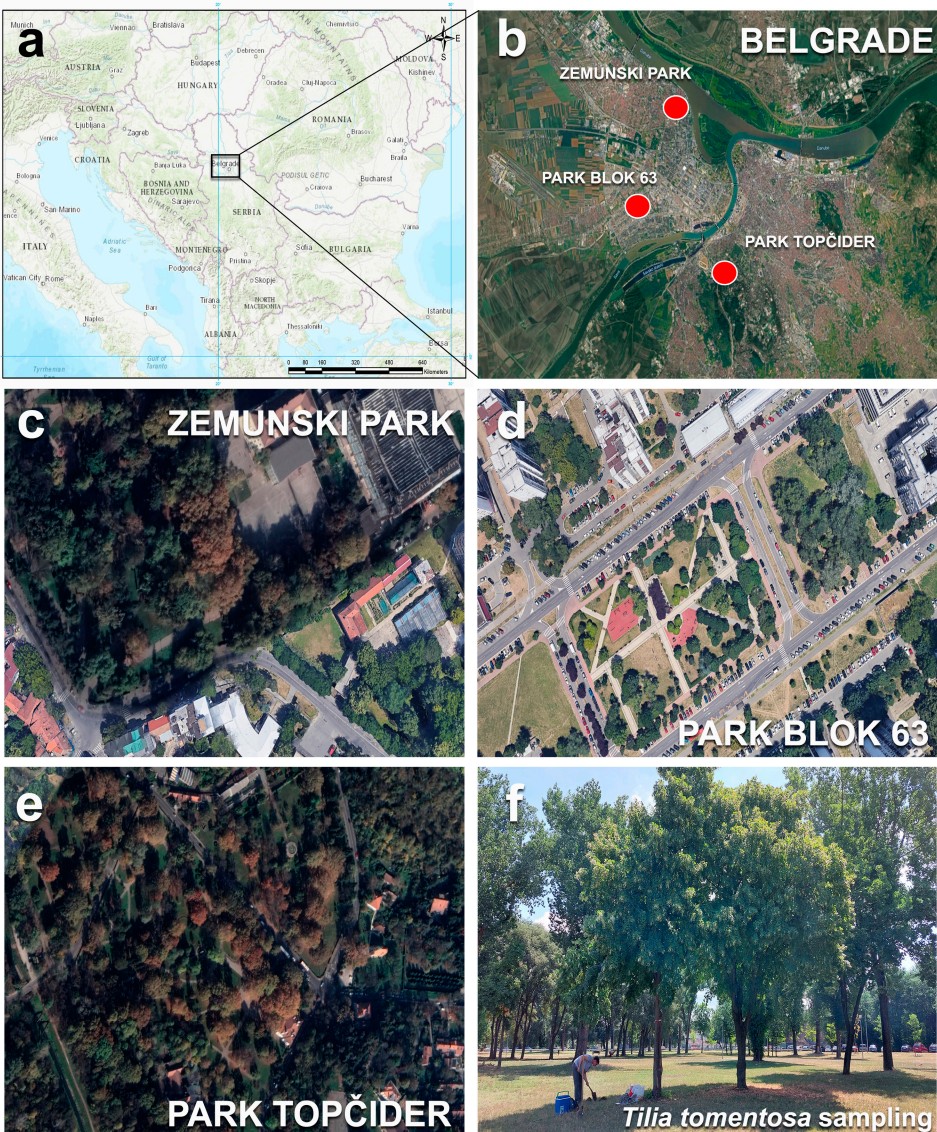

**Figure 1.** Details of the study sites in Belgrade: (**a**) Location of Belgrade on the Balkan Peninsula Penninsula (ArcGIS); (**b**) red dots represent the positions of the study sites in Belgrade; (**c**) sampling site Zemunski Park (ZP); (**d**) sampling site Park Blok 63 (PB63); (**e**) sampling site Park Topčider (PT); (**f**) *T. tomentosa* sampling. The satellite images in panels b to e are obtained from GoogleEarth (https://earth.google.com/web/@44.84419607,20.4337598,69.89526192a,40292.59140822d,35y,0h,0t,0r, accessed on 26 August 2023).

At each of the three sampling locations, 15 trees with an average age of 20–30 years were randomly chosen for *T. tomentosa* plant material and associated soil sampling. The data on the trees' ages were obtained from the Greenery-Belgrade Public Utility Enterprise (JKP Zelenilo-Beograd) (http://gispublic.zelenilo.rs/giszppublic/map, accessed on

5 May 2021). Bark, leaf, and flower samples were collected from each tree (50 g per sample). Aboveground plant parts were sampled at the same height and at all four cardinal compass points. During sampling, care was taken to avoid selecting samples with imperfections/damage, such as insect infestation or leaf damage (e.g., chlorosis or necrosis). A total of 15 bark samples, 15 leaf samples, and 15 flower samples were collected from each site. In the laboratory, the bark samples were washed with tap and distilled water, and then all the collected samples of plant material were dried separately to a constant weight and stored in plastic bags until they were analysed. Before analysis, the samples were crushed and sieved using a 2 mm aperture stainless steel sieve. Soil sampling was carried out in the root zone of each tree (250 g per tree) at a depth of 0–10 cm using stainless steel tools. Small stones, plants, and other foreign objects were removed from the soil samples by hand. In the laboratory, each soil sample was air-dried at room temperature to a constant mass and sieved through a 2 mm aperture stainless steel sieve. A total of 15 soil samples were collected from each site.

### 2.2. Soil and Plant Analysis

The prepared samples were then used for analysis of the PTE content (As, Cr, Cu, Ni, Pb, Sr, and Zn). Each sample of soil and plant material (root, leaf, and flower) was analysed in 3 replicates with 45 measurements per location ($n = 45$). The soil (0.5 g) and plant samples (0.3 g) were transferred to Teflon (iPrep) vessels. Soil mineralization was conducted through wet digestion, using the USEPA 3050B method [42], while the plant samples were digested using the USEPA 3052 method [43] in a CEM MARS 6 Microwave Acceleration Reaction System microwave oven (Matthews, NC, USA). The final extracts were filtered and transferred into 50 mL volumetric flasks and diluted to the mark with deionized water. The concentrations of PTEs in the plant and soil samples were determined by inductively coupled plasma optic emission spectroscopy (ICP-OES, Spectro Genesis, Spectro-Analytical Instruments GmbH, Kleve, Germany).

All reagents used for plant digestion were of analytical grade (Merck, Darmstadt, Germany). The ICP multi-element standard stock solutions (concentration of elements, 1000 mg L$^{-1}$ in diluted nitric acid) used to prepare standard solutions for ICP-OES analysis were also obtained from Merck. Quality control and quality assurance of the analytical data were performed by using standard reference material for leaves (beech leaves—BCR-100) and certified reference material for soil (clay—ERM-CC141), (Institute for Reference Materials and Measurements (IRMM), Geel, Belgium), as well as by analysing both reagent blanks and replicates. The recovery values were within 95.0%–110.0% for plant material and within 85.3%–106% for soil, demonstrating that the measured and certified values were in good agreement. The detection limits for the analysed elements in the soil samples were as follows (mg kg$^{-1}$): As—0.0229, Cr—0.0011, Cu—0.0014, Ni—0.0021, Pb—0.005, Sr—0.0008, and Zn—0.0023.

### 2.3. Bioconcentration and Translocation Factors

The bioconcentration patterns were calculated using the ratios between the soil and the vegetative parts (roots, leaves, and flowers), while the translocation patterns in the different plant parts were calculated using the ratios between the aboveground parts (leaves and flowers) and the underground part (roots).

The bioconcentration factor (BCF) was calculated using Equations (1) and (2), and the translocation factor (TF) was calculated using Equation (3) [44]:

$$BCF = \frac{C_{root}}{C_{soil}} \tag{1}$$

$$BCF = \frac{C_{leaf/flower}}{C_{soil}} \tag{2}$$

where $C_{root}$ represents the content (mg kg$^{-1}$) of the selected element in roots, $C_{soil}$ is the content of the same element in soil, and $C_{leaf/flower}$ is the content of the selected element in leaves and flowers;

$$TF = \frac{C_{leaf/flower}}{C_{root}} \qquad (3)$$

### 2.4. Health Risk Indices

An assessment of the effects of the selected PTEs present in *T. tomentosa*, used as a medicinal plant, on human health (children and adults) was conducted in accordance with USEPA recommendations [45]. The following parameters were used to evaluate the risk to human health posed by PTEs in the examined tree species: Estimated Daily Intake (EDI), Daily Intake of Metals (DIM), Total Hazard Quotient (THQ), and Carcinogenic Risk (CR).

EDI (mg kg$^{-1}$) (Equation (4)) is an estimate of the average concentrations of chemical elements ingested daily by a consumer. This parameter does not account for elements excreted through metabolic processes. For this purpose, EDI was calculated using the following formula:

$$EDI = C_{plant} \times I \qquad (4)$$

where (mg kg$^{-1}$) $C_{plant}$ is the metal content in the plant [mg kg$^{-1}$ d.w.], and I is the average daily adult intake rate (mg kg$^{-1}$). For this study, it was assumed that 3 cups of tea were consumed per day, with approximately 2 g of the plant being used for each cup (i.e., in this case, I = 0.006 kg day$^{-1}$).

DIM (mg kg$^{-1}$ day$^{-1}$) (Equation (5)) represents the ratio between EDI and body mass [3,46,47]:

$$DIM = \frac{C_{plant} \times I}{BM} \qquad (5)$$

where BM is the body mass (kg). For this study, the body mass was taken to be 80 kg for adults and 15 kg for children.

THQ (Equation (6)) represents an assessment of the health risk posed by the consumption of contaminated tea [46]:

$$THQ = \frac{DIM}{RfD} \qquad (6)$$

where RfD is the oral reference dose (mg kg$^{-1}$ day$^{-1}$).

Total Hazard Quotient (THQ) is used to estimate the potential health effects that can be expected as a result of the consumption of medicinal plants [46]. If the total THQ ($T_{THQ}$) value calculated using Equation (7) is less than 1, no adverse health effects are expected. If the value is higher than 1, protective measures must be taken because there is a potential health risk.

$$T_{THQ} = \sum_{i=1}^{n} THQ_i \qquad (7)$$

CR (Equation (8)) was calculated only for those elements with defined slope factors ($CSF_o$), such as As, Pb, and Cr (VI), and was used to estimate the probability of developing cancer as a result of exposure to PTEs. CR was calculated using the following formula [46]:

$$CR = CSF_o \times EDI \qquad (8)$$

### 2.5. Data Analysis

One-way analysis of variance (ANOVA) was performed to test the differences in PTE accumulation in soil samples and *T. tomentosa* roots, leaves, and flowers in relation to the collection site, i.e., the urban park (subsequent tests of normality were performed using the Shapiro–Wilk W test; Levene's test of homogeneity of variances showed non-significant differences for all the reported ANOVA breakdowns), and the means were separated using the Bonferroni test. Differences were assumed to be statistically significant at $p < 0.001$.

Canonical discriminant analysis (CDA) was performed to detect which PTE in different plant parts contributed most to the differences between the investigated sites. All statistical analyses were performed using the SPSS software package, Version 21, and the Statistica program package (SPSS Inc.: Chicago, IL, USA) [48].

## 3. Results and Discussion

### 3.1. Potentially Toxic Element Concentrations, Reference Values, and Regulatory Reference Values for Soil

Urban soils are a vital component of urban ecosystems and greatly influence human health due to the intense anthropogenic activity in urban habitats. With regard to this, one of the main problems identified in urban soils is the presence of traditional inorganic pollutants, e.g., heavy metals, accumulated in soils or accumulated in plants growing in polluted soils [3,7,49].

Both direct and indirect effects of soil on human health have been observed. Humans are directly exposed to contaminants in soil (e.g., PTEs) through skin contact, inhalation, ingestion, and consumption of plants grown in polluted soil. Indirect effects include interactions between soil and plants, as well as between plants and the nutritional value of plant foods (e.g., medicinal plants) and human health [4,7,50]. Therefore, it is important to evaluate the content of PTEs and potential contamination in soils where *T. tomentosa* grows.

As shown in Figure 2 and Table 1, the mean PTE concentrations in the examined soils varied greatly between urban parks. The concentration ranges of As, Cr, Cu, Ni, Pb, Sr, and Zn in the studied soils were 11.11–16.49, 60.75–72.43, 46.81–53.21, 52.37–92.76, 45.17–55.70, 91.27–170.33, and 104.74–147.03 mg kg$^{-1}$, respectively. Significant differences for most of the analysed elements were found between Zemunski Park and the other two parks ($p < 0.001$), while differences in the concentrations of As, Ni, Pb, Sr, and Zn were not detected between Park Blok 63 and Park Topčider. The highest concentrations of As, Cu, Ni, Pb, Sr, and Zn were measured in Zemunski Park, while the highest Cr content was measured in Zemunski Park and Park Blok 63.

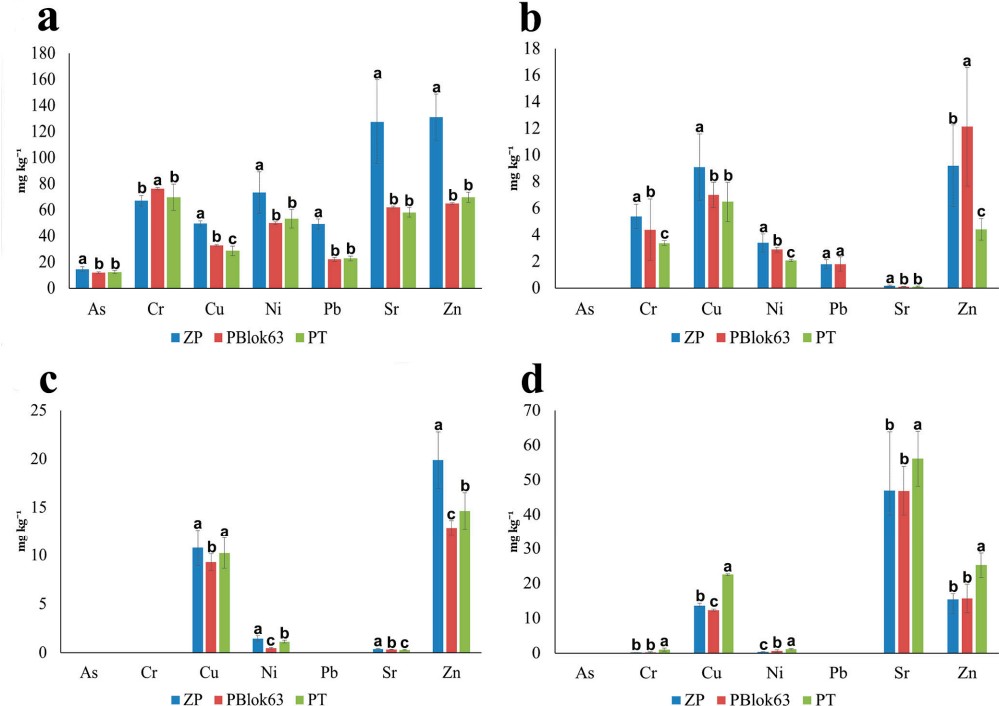

**Figure 2.** (**a**) PTEs in urban park soils; (**b**) *T. tomentosa* roots; (**c**) leaves; (**d**) flowers. Data represent the mean values (mg kg$^{-1}$) with standard deviation (M ± SD) of the samples ($n = 45$); One-way ANOVA–Bonferroni; different letters indicate significant differences between sites at $p < 0.001$.

The results obtained showed that the measured contents of PTEs in the examined soils were above the average values for worldwide soils, except for Sr, whose concentration was within the average range (87–210 mg kg$^{-1}$; [51]). The content of PTEs in the soils also exceeded the average concentrations of these elements in the soils of Serbia [52]. However, following the guidelines from Alloway [53], only the concentration of Zn fell within a range that can be critical for plant growth and development (70–400 mg kg$^{-1}$), whereas the levels of the other PTEs in the soils were below this range (Table 1). Based on Serbian legislation, all the measured PTE concentrations in the soils were below the maximum permissible concentrations (MPC), except for Ni levels, which were above the permissible value of 50 mg kg$^{-1}$ [54]. The elevated Ni content in the soils of Zemunski Park and Park Topčider, which was above the local and international reference values, indicated a potential risk posed by the consumption of *T. tomentosa* growing in these parks.

**Table 1.** Concentrations of the analysed PTEs in soils compared to reference guidelines and similar studies; the content is expressed in mg kg$^{-1}$.

| | As | Cr | Cu | Ni | Pb | Sr | Zn | Reference |
|---|---|---|---|---|---|---|---|---|
| Results of this study | 11.11–16.49 | 60.75–72.43 | 46.81–53.21 | 52.37–92.76 | 45.17–55.70 | 91.27–170.33 | 104.74–147.03 | |
| Average for worldwide soils | 4.4–8.4 | 47–51 | 13–23 | 13–26 | 22–28 | 87–210 | 45–60 | [51] |
| Average for Serbian soils | 11 | 50 | 30 | 38 | 40 | / | 48 | [52] |
| Maximum permissible concentrations (MPC) | 25 | 100 | 100 | 50 | 100 | / | 300 | [54] |
| Critical range for plants | 20–50 | 75–100 | 60–125 | >100 | 100–400 | / | 70–400 | [53] |
| Belgrade | / | 83.5–118 | 30.4–98.4 | 62.4–73.6 | 54–237 | / | 122.7–215.6 | [26] |
| Belgrade | / | 22.6–34 | 20–23.3 | 27.9–62.1 | 9.2–27.6 | 9–31.1 | 41.2–51.5 | [28] |
| Reading (UK) | / | 5.1–8.7 | 6.7–38 | 2.2–15.4 | 18.4–141 | 4.2–47.2 | 24.5–137 | |
| Aviero (PT) | / | 6–16 | 7–61 | 6–28 | 7–41 | / | 18–134 | |
| Glasgow (UK) | / | 17–131 | 24–678 | 16–53 | 41–894 | / | 64–377 | |
| Ljubljana (SI) | / | 11–33 | 20–101 | 15–43 | 39–255 | / | 81–301 | [55] |
| Sevilla (ES) | / | 19–51 | 30–86 | 21–37 | 40–265 | / | 69–171 | |
| Torino (IT) | / | 105–310 | 44–308 | 154–335 | 53–257 | / | 108–317 | |
| Uppsala (SE) | / | 12–56 | 8–180 | 7–36 | 7–211 | / | 27–247 | |

The results obtained in this study are similar to the results of previous studies investigating the urban areas of Belgrade and other European cities with a similar level of urbanization. Namely, research by Tomašević et al. [26] revealed similar values for Cu, Ni, and Zn in the urban parks of Belgrade, while the values for Cr and Pb were slightly higher than those measured in our study. These differences may be due to the selection of the parks studied in relation to the pollution sources. A study was also conducted on urban soils under linden trees in Belgrade and Reading (UK) [28]. These soils had similar levels of Cu, Ni, Pb, and Zn, but the measured Cr and Sr values were lower in comparison to those found in the current study. These differences may again be due to the differences in the parks selected for analysis in relation to the pollution sources and also to the origin of the soils (e.g., anthropogenically formed as a mixture of materials, highly transformed through mixing, importing and exporting material, and by contamination, or naturally derived from the geological substrate) [56]. Compared with the levels of PTEs in soils from other European urban parks, similar levels of Cu, Pb, and Zn were measured in Aviero (Portugal), Glasgow (United Kingdom), Ljubljana (Slovenia), Seville (Spain), Torino (Italy), and Uppsala (Sweden) by Madrid et al. [55], because these elements are mainly associated with traffic. The values of Cr and Ni measured in the soils examined in our study were generally higher than those found in the cities mentioned above, mainly due to differences in the paedogenic processes in different geological substrates [49] (Table 1).

*3.2. Accumulation of PTEs in T. tomentosa Roots, Leaves, and Flowers*

Studies have already indicated high PTE concentrations in the biomass of different medicinal plants cultivated in urban areas with traffic-related air pollution. This factor indirectly impacts PTE absorption by plants, resulting in contaminant levels above those acceptable for plant consumption [57,58].

In this study, differences in the ability of *T. tomentosa* to accumulate PTEs (As, Cr, Cu, Ni, Pb, Sr, and Zn) in its roots, leaves, and flowers in the three parks were established. Statistically significant differences for Cu and Sr levels in roots were found between trees growing in Zemunski Park and those growing the other two parks ($p < 0.001$), while there were differences in the concentrations of Cr, Ni, and Zn between all three parks (Figure 2). The highest concentrations of Cr, Cu, Ni, and Sr in roots were measured in Zemunski Park, while the highest content of Zn was measured in Park Blok 63. In leaves, there were significant differences in Ni, Sr, and Zn content between the three urban parks, while similar Cu concentrations were measured in the leaves of plants from Zemunski Park and Park Topčider. In flowers, significant differences in the Cu and Ni levels were found between the three parks, while similar concentrations of Cr, Sr, and Zn were measured in *T. tomentosa* flowers from Zemunski Park and Park Blok 63 (Figure 2). In all plant samples (roots, leaves, and flowers), the concentration of As was below the detection limit (<dl).

In the analysed plant material, most of the measured PTE concentrations were within the normal range for plants [51], with the exception of the Sr concentration in flowers. In fact, at all sampling sites, the Sr content in *T. tomentosa* flowers was in the toxic range for plants (>30 mg kg$^{-1}$; [51,59]). Exceptions were also found for roots from Zemunski Park, where the Cr levels were in the toxic range for plants (>5 mg kg$^{-1}$), and for flowers from Park Topčider, where toxic concentrations of Cu were measured (>20 mg kg$^{-1}$; [51]). These results may indicate a potential risk of using *T. tomentosa* flowers from all the examined parks for medicinal purposes, considering the toxic levels of Cu and Sr (Figure 2).

To evaluate the PTE allocation pattern between the soil and the aerial parts of *T. tomentosa*, bioaccumulation (BCF) and translocation (TF) factors were used. For a plant to be classed as a stabiliser, the BCF in roots can be higher or lower than 1, but the TF must always be lower than 1. On the other hand, plants with both BCF and TF > 1 are suitable for phytoextraction [44]. Plants with both BCF and TF below 1 can be classed as excluders. The presence of Cr was detected in roots and flowers, which confirmed previous data obtained for *T. cordata*, with higher amounts of trace elements (except for Zn) accumulated in roots than in other plant parts [32]. Unlike previous studies on Tilia species from Belgrade's urban parks [27,29] and urban sites with similar levels of pollution (3.25 mg kg$^{-1}$ [33]), the levels of Cr in leaves were low (<dl). In this study, the levels of this element in roots were above the normal range for all the parks and were within the toxic range for plants (>5 mg kg$^{-1}$; [51]) for Zemunski Park. In flowers, the Cr concentration ranged between 0.19 and 1.01 mg kg$^{-1}$, with maximum values measured in flowers from Park Topčider. These levels were similar to those previously measured by Shchukin et al. [33]. Both BCF and TF for Cr were lower than 1, indicating that *T. tomentosa* can be considered an excluder of Cr (Table 2).

The ranges of copper concentrations in plant material were as follows: 6.48–9.10 mg kg$^{-1}$ (roots), with the highest levels measured in samples from Zemunski Park, 9.31–10.79 mg kg$^{-1}$ (leaves), with the highest levels again measured in samples from Zemunski Park, and 12.4–22.49 mg kg$^{-1}$ (flowers), with the highest levels measured in samples from Park Topčider. The concentrations in leaves are in agreement with earlier findings [26,27], while we found higher levels of Cu in flowers in comparison to other studies (8.90 mg kg$^{-1}$, [60]). The TF values for Cu were greater than 1 in samples from all sites (Table 2). Considering that *T. tomentosa* is highly capable of translocating Cu from its roots to the leaves and flowers, consumers of this medicinal plant from Park Topčider, where this species was shown to accumulate toxic levels of Cu in its flowers, are exposed to potential health risks.

**Table 2.** Transfer of PTEs from soil to plant parts (BCF: Bioconcentration Factor) and between plant parts (TF: Translocation Factor). Values higher than 1 are in bold.

| Locality | Transfer | Cr | Cu | Ni | Pb | Sr | Zn |
|---|---|---|---|---|---|---|---|
| Zemunski Park | BCF Root | 0.08 | 0.19 | 0.05 | 0.04 | 0.00 | 0.07 |
| | BCF Leaf | 0.00 | 0.22 | 0.02 | 0.00 | 0.00 | 0.16 |
| | BCF Flower | 0.00 | 0.27 | 0.00 | 0.00 | 0.42 | 0.12 |
| | TF Leaf/Root | 0.00 | **1.23** | 0.44 | 0.00 | **2.21** | **2.32** |
| | TF Flower/Root | 0.04 | **1.62** | 0.10 | 0.00 | **260.00** | **1.83** |
| Park Blok 63 | BCF Root | 0.06 | 0.21 | 0.06 | 0.08 | 0.00 | 0.19 |
| | BCF Leaf | 0.00 | 0.29 | 0.01 | 0.00 | 0.01 | 0.20 |
| | BCF Flower | 0.00 | 0.38 | 0.01 | 0.00 | 0.76 | 0.24 |
| | TF Leaf/Root | 0.00 | **1.36** | 0.17 | 0.00 | **3.90** | **1.17** |
| | TF Flower/Root | 0.09 | **1.79** | 0.20 | 0.00 | **556.00** | **1.33** |
| Park Topčider | BCF Root | 0.05 | 0.22 | 0.04 | 0.00 | 0.00 | 0.06 |
| | BCF Leaf | 0.00 | 0.37 | 0.02 | 0.00 | 0.01 | 0.21 |
| | BCF Flower | 0.02 | 0.80 | 0.02 | 0.00 | 0.97 | 0.37 |
| | TF Leaf/Root | 0.00 | **1.73** | 0.54 | 0.00 | **2.98** | **3.52** |
| | TF Flower/Root | 0.30 | **3.68** | 0.55 | 0.00 | **573.00** | **6.00** |

Transfer from soil to root: BCF Root; Transfer from soil to leaf: BCF Leaf; Transfer from soil to flower: BCF Flower; Transfer from root to leaf: TF Leaf/Root; Transfer from root to flower: TF Flower/Root

The Ni content in *T. tomentosa* roots ranged between 2.08 mg kg$^{-1}$ in samples from Park Topčider and 3.39 mg kg$^{-1}$ in samples from Zemunski Park. In leaves, the Ni levels ranged from 0.48 to 1.46 mg kg$^{-1}$, while in flowers, the levels were between 0.33 and 1.15 mg kg$^{-1}$, with maximum concentrations measured in samples from Park Topčider for both leaves and flowers (Figure 2). The Ni levels were within the normal range in all plant parts. These findings confirmed earlier ones in Tilia species from several urban parks in Belgrade [26,27], and the measured values were found to be far lower in comparison to those found in other urban habitats [33]. The results also indicated that *T. tomentosa* can be considered an excluder of Ni, according to the given guidelines (both BCF and TF less than 1; Table 2).

The presence of lead in *T. tomentosa* plant material was measured only in roots, with concentrations ranging between <dl and 1.80 mg/g in samples from both Zemunski Park and Park Blok 63. In most plants, 90% of the total Pb accumulated in roots and was localised in the insoluble fraction consisting of cell walls and nuclei [61,62]. These values were within the normal range for plants (0.1–5 mg kg$^{-1}$, [51]), far lower than the maximum permissible levels for lead in medicinal plants, as prescribed by the World Health Organization (10 mg kg$^{-1}$; [63]). The Serbian legislation [64] only reports the maximum permitted concentration of lead in homemade tea, which is 5 mg kg$^{-1}$. As for Cu and Ni, these results indicated that *T. tomentosa* can be considered an excluder of Pb, according to both BCF and TF (both < 1; Table 2).

The concentrations of Sr in root material were within a narrow range, i.e., 0.10–0.18 mg kg$^{-1}$, with the maximum content measured in samples from Zemunski Park. Higher Sr concentrations were measured in leaves, ranging between 0.29 and 0.38 mg kg$^{-1}$, again with the maximum level measured in samples from Zemunski Park. By far, the highest levels of Sr were measured in flowers. Concentrations between 46.82 and 56.07 mg kg$^{-1}$ were recorded, with the highest ones measured in samples from Park Topčider. Plants can be contaminated with Sr via root uptake, leaf uptake, and the deposition of the contaminant on aerial parts. Only a minor part of Sr deposited on the leaves is absorbed and re-translocated to other parts of the plant, probably because of the non-mobility of Sr in the phloem [65,66]. In regard to the potential toxicity of this element, there is a limited number of studies on Sr accumulation and toxicity in plants. So far, Shacklette et al. [59] set the potentially toxic levels above 30 mg kg$^{-1}$. The TF values in relation to root–flower translocation in *T. tomentosa* are up to several hundred times higher than 1 (Table 2), meaning *T. tomentosa* is highly capable of translocating PTEs from its roots to the leaves and flowers, accumulating

toxic levels of Sr in its flowers, which therefore poses potential health risks for consumers of this medicinal plant. It was previously established that long exposure periods to high levels of Sr led to the accumulation of this metal in the aboveground parts of plants [67]. Our previous research into Sr accumulation in trees growing in urban parks revealed elevated concentrations in *A. hippocastanum* leaves (107.5 mg kg$^{-1}$) [17] and in Tilia spp. leaves (170 mg kg$^{-1}$ in Belgrade and 83.5 mg kg$^{-1}$ in Reading) [28]. Considering that the root and leaf concentrations of Sr were significantly lower compared to those in flowers, it is reasonable to assume that most Sr content in *T. tomentosa* flowers derived from airborne Sr, confirming our earlier findings of high levels of this element measured both in the leaves and in the bark of trees growing in urban parks [17]. When discussing PTEs originating from air pollution, it should be noted that PTE sources can be both PM10 and PM2.5 (particulate matter). The PTE concentrations in these particles can be a valuable indicator of air pollution because the polluting elements may be associated with falling dust. In this case, data from the Serbian Environmental Protection Agency (2022) [68] gathered from the state-run monitoring stations closest to the study sites for the 2021 sampling period showed elevated levels of PM10, higher than the annual limit values (40 µg m$^{-3}$), in the vicinity of Zemunski park (ZP) and Park Topčider (PT), while the concentrations of PM2.5 fell within the permissible levels. Likewise, the maximum daily limit value for PM10 of 50 µg m$^{-3}$ was exceeded at all sites. In terms of the concentrations of PTEs in PM10, the data showed the following: the mean annual limit concentrations of As, Ni, and Pb were within the permissible limits, while the maximum daily limit concentrations of As and Ni largely exceeded the target mean annual values, which did not affect the measured concentrations of these elements in the aerial parts of the examined species (Supplementary Materials, Tables S1 and S2).

The concentrations of Zn in plant material were as follows: 4.41–12.13 mg kg$^{-1}$ (roots), with the maximum concentration measured in samples from Park Blok 63, 12.89–19.87 mg kg$^{-1}$ (leaves), with the maximum value again measured in samples from Park Blok 63, and 15.36–25.21 mg kg$^{-1}$ (flowers), with the maximum level measured in samples from Park Topčider. In leaves, the Zn levels were in line with those reported in previous studies of Tilia species from Belgrade's urban parks [26,27]. In flowers, the Zn contents were higher than in roots and leaves, but within the normal range for plants. The Zinc concentrations in flowers were higher compared to those previously reported for Tilia species from urban habitats (8.60 mg kg$^{-1}$, [60]). The BCF values of Zn were less than 1, while the TF values were greater than 1 at all the sites, thus indicating the high ability of *T. tomentosa* to translocate Zn as a valuable essential element from its roots to the leaves and flowers.

To determine the differences between roots, leaves, and flowers of *T. tomentosa* in the accumulated content of PTEs (the contents of As, Cr, and Pb were not included in the analysis because these elements accumulated below their detection limits), canonical discriminant analysis (CDA) was applied, and the results are shown in Figure 3. The differences in the content of PTEs in roots, leaves, and flowers were determined on the basis of the first discriminant axis (CD1), which explained 63.70% of the variability and was largely determined by the content of Cu, Ni, and Sr, while the second component (CD 2) explained 36.30% of the variability and was determined by the content of Ni and Zn.

### 3.3. Potential Health Risks Arising from Using *T. tomentosa* Leaves and Flowers for Medicinal Purposes

Despite having a reputation for being natural and therefore harmless, plant remedies can certainly have negative effects. Namely, many medicinal plants and preparations made from them can present a health risk due to the presence of toxic elements such as Pb, Cd, Al, and Hg and other elements like Cr, which are hazardous to humans, depending on their oxidation states and on whether they are present in high concentrations [69,70]. The level of essential elements in plants is affected by the geochemical characteristics of the soil and by the ability of plants to selectively accumulate some of these elements [71]. For example, copper and zinc are essential micronutrients and have a variety of biochemical functions

in living organisms, but these elements can be toxic when taken in excess. Therefore, herbal medicines that are contaminated with PTEs have become a global threat to human beings, especially when the PTE levels are above the known threshold concentrations [70]. Thus, a good quality control is a must for medicinal plants in order to protect consumers from contamination.

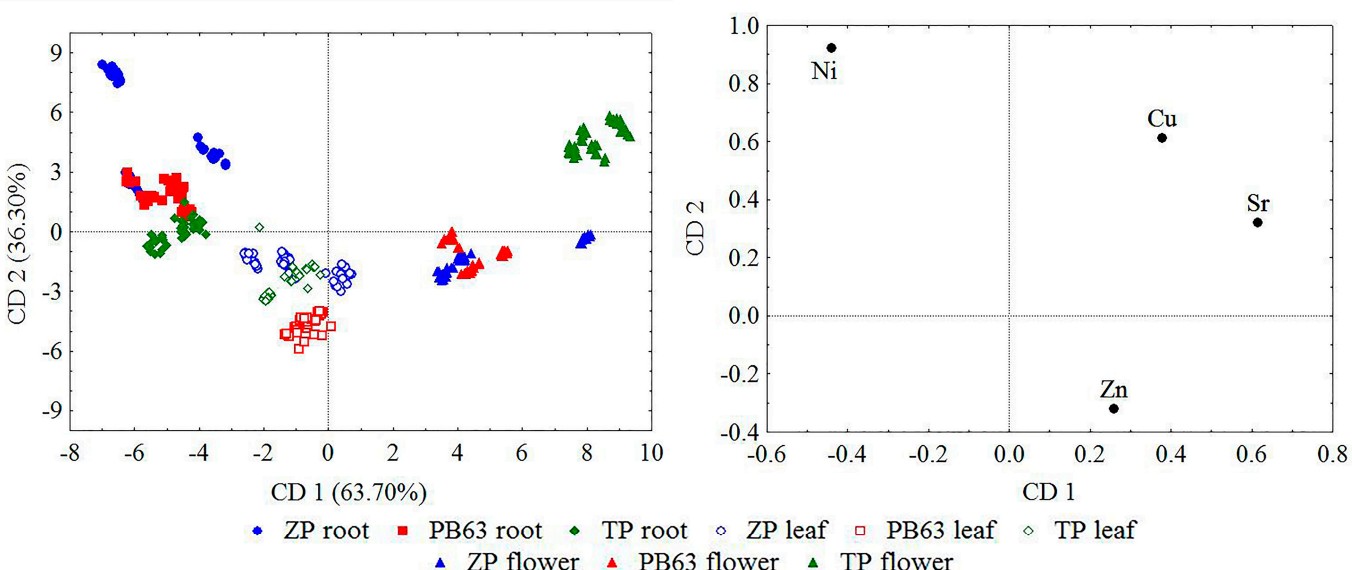

**Figure 3.** CDA for selected elements in *T. tomentosa* roots, leaves, and flowers.

Since the Middle Ages, the flowers of the Tilia tree have been primarily used as a diaphoretic to promote sweating. They have also been used for a variety of other medicinal purposes in phytotherapy, including as an expectorant, diuretic, antispasmodic, stomachic, and sedative [35,38]. In addition, the flowers have been used traditionally for the treatment of flu, coughs, migraines, nervous tension, ingestion, various types of spasms, liver and gall bladder disorders, diarrhoea, and elevated arterial pressure associated with arteriosclerosis. *Tilia* flowers are an ingredient in common cold and antitussive preparations, as well as in urological and sedative drugs. In paediatric medicine, they are included with those of several other species as a diaphoretic component in a tea used to treat influenza [35,72]. Several pharmacologically active compounds from *T. tomentosa* have been isolated [34,73].

In urban areas, there is a correlation between the use of medicinal plants and the socio-economic level of a population. Medicinal plants tend to be used more by low-income classes, either because other alternatives are too expensive or because of the persisting tradition of folk medicine. However, phytomedicines are increasingly being used in more developed countries too, as the desire for natural products becomes more and more popular. Labelled as natural, people do not realise the possible risks associated with using these products [74]. The most popular way an urban population obtains medicinal plants is by harvesting them in open green spaces or by cultivating them in urban gardens. However, the consumption of medicinal plant materials harvested under urban conditions can result in various pollutants entering the human body, primarily PTEs, implying the need for in situ monitoring the accumulation of potentially toxic chemical elements in such plants [28,74–76].

The EDI of PTEs through the consumption of linden tea by people is reported in Table 3. The EDI of the studied PTEs (As and Pb were not included due to their low concentrations, <dl) was calculated considering the consumption of three cups of tea per day with approximately 2 g of the plant being used for each cup. The European Food Safety Authority (EFSA) proposed an Adequate Intake (AI) for Cu of 1.6 mg/day for adult men, 1.3 mg day$^{-1}$ for adult women, and within the range from 0.7 mg day$^{-1}$ to 1.1 mg day$^{-1}$ for children [77]. Thus, the results of this study indicated the lack of potential health risks

due to excessive copper intake from using linden tea produced with material from the studied parks. Also, the Average Requirements (ARs) for dietary Zn necessary to meet physiological requirements are estimated between 7.5 and 12.7 mg day$^{-1}$ for women and 9.4 to 16.3 mg day$^{-1}$ for men, while the ARs for infants and children range from 2.4 to 11.8 mg day$^{-1}$ [78]. The results obtained in this study revealed a Zn content much lower than the EDI, meaning a lack of health risks from excessive Zn intake.

**Table 3.** Estimated Daily Intake (EDI) of the studied PTEs (mg day$^{-1}$) following the consumption of *T. tomentosa* flowers and leaves. The values higher than the Provisional Tolerable Daily Intake (PTDI) are in bold.

| Locality/Element | Flowers | | | | | Leaves | | | |
|---|---|---|---|---|---|---|---|---|---|
| | Cr | Cu | Ni | Sr | Zn | Cu | Ni | Sr | Zn |
| Zemunski park | 0.001 | 0.081 | 0.002 | 0.281 | 0.092 | 0.065 | **0.009** | 0.002 | 0.119 |
| Park Blok 63 | 0.001 | 0.074 | 0.003 | 0.281 | 0.094 | 0.056 | 0.003 | 0.002 | 0.077 |
| Park Topčider | 0.006 | 0.135 | **0.007** | 0.336 | 0.151 | 0.061 | **0.007** | 0.002 | 0.088 |

The Provisional Tolerable Daily Intake (PTDI) is 0.03 for Cr, 0.5 for Cu, 0.005 for Ni, and 1 for Zn [79–81]. Compared to these values, the EDI of Ni from the consumption of both flowers and leaves from Park Topčider was above the PTDI, which could cause potential health problems. The same was true for Ni in leaves from Zemunski Park (Table 3).

As regards the DIM measurements for the examined PTEs, the values for in flowers was not detected for As, Cr, Ni, and Pb, from the consumption of flowers were below the detection limit, while as were those for DIM in leaves was not detected for As, Cr, Pb, and Sr from the consumption of leaves. The values obtained are shown in Table 4. The calculated DIM values were generally higher for flowers than for leaves and higher for children than for adults.

**Table 4.** Daily Intake of Metals (DIM) from *T. tomentosa* flowers and leaves, for adults and children (in mg kg$^{-1}$ day$^{-1}$).

| Flowers | | | | | | |
|---|---|---|---|---|---|---|
| | Adults | | | Children | | |
| Locality/Element | Cu | Sr | Zn | Cu | Sr | Zn |
| Zemunski Park | 0.001 | 0.004 | 0.001 | 0.005 | 0.019 | 0.006 |
| Park Blok 63 | 0.001 | 0.004 | 0.001 | 0.005 | 0.019 | 0.006 |
| Park Topčider | 0.002 | 0.005 | 0.002 | 0.009 | 0.022 | 0.010 |
| Leaves | | | | | | |
| | Adults | | | Children | | |
| Locality/Element | Cu | Ni | Zn | Cu | Ni | Zn |
| Zemunski park | 0.001 | 0.000 | 0.002 | 0.004 | 0.001 | 0.008 |
| Park Blok 63 | 0.001 | 0.000 | 0.001 | 0.004 | 0.000 | 0.005 |
| Park Topčider | 0.001 | 0.000 | 0.001 | 0.004 | 0.000 | 0.006 |

THQ represents an assessment of the non-carcinogenic health risk posed by the consumption of potentially contaminated teas made from linden flowers and leaves. Its calculation is based on EDI and DIM. Generally, the individual THQ values and total values (T$_{THQ}$) for flowers were higher than those for leaves (Table 5). The results showed the lack of a significant non-carcinogenic risk for adults and children from the PTEs studied, as the THQ and THRI values were lower than 1 for all the elements and localities [46,82].

**Table 5.** Total Hazard Quotient (THQ) for the studied elements and Total THQ ($T_{THQ}$) calculated for both *T. tomentosa* flowers and leaves, for adults and children (in mg kg$^{-1}$ day$^{-1}$).

| | | | | | | | | | | | | |
|---|---|---|---|---|---|---|---|---|---|---|---|---|
| **Flowers** | | | | | | | | | | | | |
| | **Adults** | | | | | | **Children** | | | | | |
| **Locality/Element** | **Cr** | **Cu** | **Ni** | **Sr** | **Zn** | **$T_{THQ}$** | **Cr** | **Cu** | **Ni** | **Sr** | **Zn** | **$T_{THQ}$** |
| Zemunski Park | 0.005 | 0.029 | 0.001 | 0.007 | 0.004 | 0.056 | 0.025 | 0.135 | 0.007 | 0.031 | 0.004 | 0.245 |
| Park Blok 63 | 0.008 | 0.026 | 0.002 | 0.007 | 0.004 | 0.062 | 0.036 | 0.123 | 0.011 | 0.031 | 0.004 | 0.274 |
| Park Topčider | 0.029 | 0.048 | 0.005 | 0.008 | 0.007 | 0.114 | 0.134 | 0.225 | 0.023 | 0.037 | 0.007 | 0.506 |
| **Leaves** | | | | | | | | | | | | |
| | **Adults** | | | | | | **Children** | | | | | |
| **Locality/Element** | **Cr** | **Cu** | **Ni** | **Sr** | **Zn** | **$T_{THQ}$** | **Cr** | **Cu** | **Ni** | **Sr** | **Zn** | **$T_{THQ}$** |
| Zemunski Park | 0.00 | 0.023 | 0.006 | 0.00 | 0.006 | 0.058 | 0.00 | 0.108 | 0.029 | 0.00 | 0.006 | 0.250 |
| Park Blok 63 | 0.00 | 0.020 | 0.002 | 0.00 | 0.004 | 0.046 | 0.00 | 0.093 | 0.010 | 0.00 | 0.004 | 0.202 |
| Park Topčider | 0.00 | 0.022 | 0.005 | 0.00 | 0.004 | 0.051 | 0.00 | 0.102 | 0.023 | 0.00 | 0.004 | 0.223 |

CR was calculated only for Cr in flowers, since As and Pb were below the detection limits and there was no slope factor to use for the other elements.

The CR values for Cr in flowers from Zemunski Park ($5.591 \times 10^{-4}$) and Park Blok 63 ($8.061 \times 10^{-4}$) were within the USEPA 'acceptable' range ($10^{-4}$–$10^{-6}$; [83]), indicating that no carcinogenic risks would be expected in residential areas, while the CR values for Cr in flowers from Park Topčider were above the USEPA limit ($3.021 \times 10^{-3}$) (Figure 4). These results indicated possible adverse effects on human health and a carcinogenic risk arising from the ingestion of flowers of *T. tomentosa* from Park Topčider due to their Cr content.

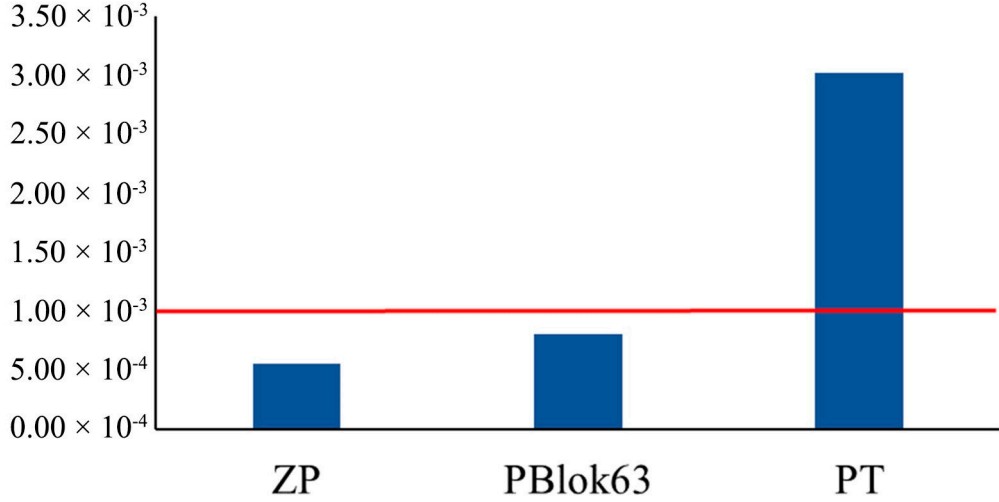

**Figure 4.** The Carcinogenic Risk (CR) for chromium in *T. tomentosa* flowers, with the 'acceptable' limit marked in red.

## 4. Conclusions

Due to the reduction in natural forests, urbanization generates changes which impact the traditional use of medicinal plants. The impact that the urban environment has on the accumulation of potentially toxic elements in medicinal plants was investigated in this study, in particular in the common urban tree *T. tomentosa* from three urban parks situated in residential areas of Belgrade, Serbia. The valorisation of commonly used linden plant parts (leaves and flowers) is challenging due to their uptake of PTEs, which presents a potential risk for their uses and consumption. Therefore, this research focused on the PTE content in soils, their accumulation and translocation patterns in *T. tomentosa* plants,

and the potential health risks arising from using this species grown in urban parks as a medicinal plant.

The results obtained showed that the measured concentrations of PTEs in the examined urban soils were above the reference values, except for Sr. However, all the measured PTE levels in soil were below the local and international reference values, except for Ni in Zemunski Park and Park Topčider, indicating a potential risk of Ni accumulation in the *T. tomentosa* trees growing in these parks. Most PTE concentrations measured in the aerial parts of *T. tomentosa* were within the normal range for plants, with the exception of Cu, for which toxic levels were measured in flowers from Park Topčider, and of Sr, with toxic levels measured in flowers from all the examined parks. The high ability of *T. tomentosa* to translocate PTEs from its roots to the leaves and flowers (TF much higher than 1) and the toxic levels of Sr in the flowers indicated a potential health risks for consumers of this medicinal plant.

In regard to the potential human health risk, based on the Estimated Daily Intake, it can be concluded that the consumption of flowers and leaves from Park Topčider and Zemunski Park would pose a potential health risk from excessive Ni intake, particularly for children, due to Ni levels above the Provisional Tolerable Daily Intake value. In addition, the Carcinogenic Risk calculated for Cr was above the USEPA limit ($3.021 \times 10^{-3}$), indicating possible adverse effects on human health and a carcinogenic risk arising from ingesting the flowers of *T. tomentosa* from Park Topčider.

In conclusion, the examined medicinal plant material of *T. tomentosa* harvested under urban conditions can be a source of PTEs (Cr, Cu, Ni, and Sr) for the human body and the cause of potential health issues, which implies the need to monitor PTE accumulation, in particular in those plant parts which are commonly used for medicinal purposes.

**Supplementary Materials:** The following supporting information can be downloaded at https://www.mdpi.com/article/10.3390/f14112204/s1, Table S1. Mean annual content and maximum daily content of $PM_{10}$ and $PM_{2.5}$ particulate matter from the nearest monitoring stations (values expressed in μg $m^{-3}$); Table S2. Mean annual content and maximum daily content of As, Ni, and Pb in $PM_{10}$ and $PM_{2.5}$ particulate matter from the nearest monitoring stations (values expressed in ng $m^{-3}$).

**Author Contributions:** Conceptualization, M.M. (Miroslava Mitrović), O.K. and P.P.; formal analysis, O.K., Z.M. and M.M. (Milica Marković); investigation, O.K., N.R. and D.S.; methodology, S.J.; visualization, M.M. (Miroslava Mitrović) and P.P.; writing—original draft, M.M. (Miroslava Mitrović), O.K. and P.P.; writing—review and editing, P.P. and M.M. (Miroslava Mitrović); funding acquisition, P.P. All authors have read and agreed to the published version of the manuscript.

**Funding:** This work was supported by the Ministry of Science, Technological Development and Innovation of the Republic of Serbia [grant no. 451-03-47/2023-01/200007].

**Data Availability Statement:** The data presented in this study are available on request from the corresponding author.

**Conflicts of Interest:** The authors declare no conflict of interest.

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
