# Peer review of "Bioaccumulation of Potentially Toxic Elements in Tilia tomentosa Moench Trees from Urban Parks and Potential Health Risks from Using Leaves and Flowers for Medicinal Purposes"

_forests, doi:10.3390/f14112204_

Round 1
Reviewer 1 Report
Comments and Suggestions for Authors
the structure of the MS is complete and the figures and tables are relevant. it is well well-written MS
the better citation will help the author provide better and more accurate information regarding plants with potential element accumulation and translocation patterns (As, Cr, Cu, Ni, Pb, Sr, and Zn)
Comments on the Quality of English Languagethe structure of the MS is complete and the figures and tables are relevant. it is well well-written MS
the better citation will help the author provide better and more accurate information regarding plants with potential element accumulation and translocation patterns (As, Cr, Cu, Ni, Pb, Sr, and Zn)
Author Response
Review #1 Report Form
the structure of the MS is complete and the figures and tables are relevant. it is well well-written MS
the better citation will help the author provide better and more accurate information regarding plants with potential element accumulation and translocation patterns (As, Cr, Cu, Ni, Pb, Sr, and Zn)
Thank you for your valuable recommendation.
In our opinion, we have referred to adequate literature sources. Suggestions on additional literature sources are very welcome.
In the revised version, we have taken on board the Editor’s suggestions related to the References section and revised it accordingly.
Reviewer 2 Report
Comments and Suggestions for Authors
Monitoring variability in heavy metal concentrations in city greenspace is needed incredibly potentially toxic element (PTE) contamination in medicinal plants, particularly those growing in urban environments. Assess the potential health risks of consuming leaves and flowers based on calculations of the Estimated Daily Intake (EDI), Daily Intake of Metals (DIM), Health Risk Index (HRI), and Carcinogenic Risk (CR) is vital to residents of cities potentially using plants for medical purposes, those collected near the place of residence.
The problem is of broad international interest. The subject of the presented paper is suitable for publication profile for "Forests". The title of the manuscript reflects the content. The manuscript is sustained by relevant literature. It is clear and well-structured. The methodology chapter is detailed, described, and supported by the literature. The results and discussions are appropriately described and presented. The tables and figures contain valid data. This work is interesting and based on a great deal of research. Measurements of PM10 and PM2.5 particulate matter from the environmental sampling period would be a valuable indicator of air pollution because the metals tested may be associated with falling dust. It might be possible to provide the reader with data from the nearest state or commercial monitoring sites in the neighbourhood of the study sites in Belgrade and to discuss this aspect as well.
Author Response
Review #2 Report Form
Monitoring variability in heavy metal concentrations in city greenspace is needed incredibly potentially toxic element (PTE) contamination in medicinal plants, particularly those growing in urban environments. Assess the potential health risks of consuming leaves and flowers based on calculations of the Estimated Daily Intake (EDI), Daily Intake of Metals (DIM), Health Risk Index (HRI), and Carcinogenic Risk (CR) is vital to residents of cities potentially using plants for medical purposes, those collected near the place of residence.
The problem is of broad international interest. The subject of the presented paper is suitable for publication profile for "Forests". The title of the manuscript reflects the content. The manuscript is sustained by relevant literature. It is clear and well-structured. The methodology chapter is detailed, described, and supported by the literature. The results and discussions are appropriately described and presented. The tables and figures contain valid data. This work is interesting and based on a great deal of research. Measurements of PM10 and PM2.5 particulate matter from the environmental sampling period would be a valuable indicator of air pollution because the metals tested may be associated with falling dust. It might be possible to provide the reader with data from the nearest state or commercial monitoring sites in the neighbourhood of the study sites in Belgrade and to discuss this aspect as well.
Thank you for valuable recommendation.
In the revised varsion this issue has been addressed as follows: When discussing PTEs originating from air pollution, it should be noted that PTEs source can be both PM10 and PM2.5 (particulate matter). Their concentrations in these particles can be a valuable indicator of air pollution because the polluting elements may be associated with falling dust. In this case, data from the Serbian Environmental Protection Agency (2022) [68] gathered from those state-run monitoring stations closest to the study sites for the 2021 sampling period showed elevated levels of PM10, higher than the annual limit values (40 μg m-3) in the vicinity of Zemunski park (ZP) and Park Topčider (PT), while concentrations of PM2.5 fell within the category of permissible levels. Likewise, the maximum daily limit value for PM10 of 50 μg m-3 was exceeded at all the sites. In terms of concentrations of PTEs in PM10, data showed the following: the mean annual limit values of As, Ni, and Pb concentrations were within the permissible limits, while the maximum daily limit values for As and Ni concentrations exceeded the target mean annual values many times over, which did not affect the measured concentrations of these elements in the aerial parts of the examined species (Supplementary material, Tables S1 and S2).
Reviewer 3 Report
Comments and Suggestions for Authors
The manuscript of Miroslava Mitrović et al. present results regarding the accumulation of trace elements in different parts of silver linden (Tilia tomentosa Moench) plant that are sampled in Belgrade’s urban parks. The translocation patterns of these trace elements were also presented and discussed. The overall design of the experiment is well-grounded, and the results are presented and discussed in a systematic manner. This manuscript, in my opinion, is of sufficient interest for publication on Forests, after a revision. There are a few issues to address.
(1) Line 90-91; Provide more description of your sampling plots and/or sites. The sample was collected during what season, climate conditions, and what is the humidity level during the time of sampling? Provide this data.
(2) Line 97; what methods are being used to confirm the tree's age? Provide the details.
(3) Lines 99-101; please indicate clearly the number of samplings for root, leaf and flower per silver linden tree. A lot of considerations on the samples need to be accurately investigated and discussed. For instance, if only one flower was sampled per tree for the analysis, this might lead to inconclusive results.
(4) Please describe how the samples (root, leaf and flower) were kept prior to the analysis.
(5) Fig. 2; The standard deviation for some of the results is relatively high with the variation of > 10% (e.g., Zn content in leaf). Provide explanations.
Author Response
Review # 3 Report Form
The manuscript of Miroslava Mitrović et al. present results regarding the accumulation of trace elements in different parts of silver linden (Tilia tomentosa Moench) plant that are sampled in Belgrade’s urban parks. The translocation patterns of these trace elements were also presented and discussed. The overall design of the experiment is well-grounded, and the results are presented and discussed in a systematic manner. This manuscript, in my opinion, is of sufficient interest for publication on Forests, after a revision. There are a few issues to address.
(1) Line 90-91; Provide more description of your sampling plots and/or sites. The sample was collected during what season, climate conditions, and what is the humidity level during the time of sampling? Provide this data.
Thank you for valuable recommendation.
In the revised manuscript, we have provided the required information on sampling sites, season, and climatic conditions during the sampling period.
(2) Line 97; what methods are being used to confirm the tree's age? Provide the details.
In the revised version, the required information has been added to the text as follows: Data on the trees’ ages was obtained from the Greenery-Belgrade Public Utility Enterprise (JKP Zelenilo-Beograd) (http://gispublic.zelenilo.rs/giszppublic/Map).
(3) Lines 99-101; please indicate clearly the number of samplings for root, leaf and flower per silver linden tree. A lot of considerations on the samples need to be accurately investigated and discussed. For instance, if only one flower was sampled per tree for the analysis, this might lead to inconclusive results.
Indeed, we agree with your remark.
In the revised version, all the required information has been provided in the Material and methods section.
(4) Please describe how the samples (root, leaf and flower) were kept prior to the analysis.
In the revised manuscript, how samples were kept has been clearly defined.
(5) Fig. 2; The standard deviation for some of the results is relatively high with the variation of > 10% (e.g., Zn content in leaf). Provide explanations.
We agree. In the revised version, in the Material and methods section, a detailed description of the sampling procedure has been provided. It is clearly stated that measurements and analyses were conducted on material sampled from 15 individual silver linden trees, which may in itself be the reason for greater variation in the concentrations of some of the analysed elements. Standard deviation > 10 % in the case of Zn confirms earlier findings by researchers of natural variation in Zn content within plant species (cited in Chen et al., 2018).
Chen ZR, Kuang L, Gao YQ, Wang YL, Salt DE, Chao DY. AtHMA4 Drives Natural Variation in Leaf Zn Concentration of Arabidopsis thaliana. Front Plant Sci. 2018 Mar 1;9:270. doi: 10.3389/fpls.2018.00270.
Round 2
Reviewer 3 Report
Comments and Suggestions for Authors
The authors have made the necessary revisions as suggested by the reviewers. The manuscript at the current stage is suitable for publication in Forests.
Author Response
Review # 3 Report Form
The authors have made the necessary revisions as suggested by the reviewers. The manuscript at the current stage is suitable for publication in Forests.
Thank you for the valuable recommendation.